# Suicidal Attempts among Secondary School-Going Adolescents in Kilimanjaro Region, Northern Tanzania

**DOI:** 10.3390/bs13040288

**Published:** 2023-03-28

**Authors:** Jackline Shirima, Lisbeth Mhando, Rehema Mavura, Innocent B. Mboya, James S. Ngocho

**Affiliations:** 1Department of Epidemiology and Applied Biostatistics, Institute of Public Health, Kilimanjaro Christian Medical University College, Moshi P.O. Box 2240, Tanzania; jacklinethadei@gmail.com (J.S.); james.ngocho@kcmuco.ac.tz (J.S.N.); 2Department of Behavioral and Social Sciences, Institute of Public Health, Kilimanjaro Christian Medical University College, Moshi P.O. Box 2240, Tanzania; lismhando@gmail.com; 3Department of Community Health, Kilimanjaro Christian Medical Center, Moshi P.O. Box 3010, Tanzania; reyahmed1994@gmail.com; 4Department of Translational Medicine, Lund University, 214 28 Malmö, Sweden

**Keywords:** suicide attempt, adolescents, global school-based student health survey, regional school-based student health survey, Kilimanjaro, Tanzania

## Abstract

Suicide attempts among adolescents are common and can lead to death. The study aimed to determine the prevalence and factors associated with suicide attempts among secondary school-going adolescents in the Kilimanjaro region, northern Tanzania. The study used data from two repeated regional school-based student health surveys (RSHS), conducted in 2019 (Survey 1) and 2022 (Survey 2). Data were analyzed for secondary school students aged 13 to 17 years from four districts of the Kilimanjaro region. The study included 4188 secondary school-going adolescents: 3182 in Survey 1 and 1006 in Survey 2. The mean age in Survey 1 was 14 years and the median age in Survey 2 was 17 years (*p* < 0.001). The overall prevalence of suicide attempts was 3.3% (3.0% in Survey 1 and 4.2% in Survey 2). Female adolescents had higher odds of suicide attempts (aOR = 3.0, 95% CI 1.2–5.5), as did those who felt lonely (aOR = 2.0, 95% CI 1.0–3.6), had ever been worried (aOR = 1.9, 95% CI 1.0–3.5), or had ever been bullied (aOR = 2.2, 95% CI 1.2–4.1). Suicidal attempts are prevalent among secondary school-going adolescents in the Kilimanjaro region, northern Tanzania. In-school programs should be established to prevent such attempts.

## 1. Introduction

Suicide among adolescents is a major public health concern worldwide [1]. It was the fourth leading cause of death among adolescents aged 15–19 years [2]. Each year, nearly 46,000 adolescents die by suicide [3]. However, 77% of all suicides among adolescents occur in low- and middle-income countries (LMICs), where most of the world’s population resides [4]. An increase in the mean suicide rate among adolescents 15–19 years has been observed at a rate of 7.4 per 100,000 [5]. Suicide behaviors include suicidal ideation (wanting to die by suicide), suicidal planning (preparing to die by suicide), suicidal attempt (making an attempt to die by suicide), and suicide itself (actually dying by suicide) [6]. Attempted suicide is a major risk factor for mortality caused by suicide in the general population, which has a negative impact on families, communities, and the entire country [7].

There are variations in suicide attempts across surveys in LMICs. For instance, the prevalence of suicide attempts was 16.6% in the 2015 Guatemala global school-based health survey (GSHS): 12.2% in boys and 20.2% in girls [8]. In South Africa, suicide attempts, suicide planning, and suicidal ideation have been reported at 3.2%, 5.8%, and 7.2%, respectively [9]. While in Malawi, a higher prevalence (12.9%) of suicide attempts among school-going adolescents was reported, with 7.2% of adolescents reporting one attempt and 5.9% of adolescents reporting making two or more attempts [10]. Few studies on adolescent suicide attempts in Tanzania show an increase in suicidal attempts from 11.1% to 24.8% [11,12] which has been linked to serious mental health crises and substance abuse. According to Tanzania’s 2014 GSHS report, 13.9% of students seriously considered suicide, 9.5% of students made a suicide plan, and 11.5% of students reported attempted suicide [13]. The feeling of loneliness, bullying, food insecurity, substance abuse, female sex, physical attack, and sexual intercourse are some of the drivers of suicidal attempts among in-school adolescents [8,14,15,16]. Physical or sexual abuse, stressful life events, mood disorders, anxiety, interpersonal violence, mental problems, low social economic status, having no close friends, and family suicide history are among the psychosocial factors that influence suicidal attempts in adolescents [16,17].

The GSHS conducted to measure risk behaviors helps in determining the burden of suicidal attempts among secondary school adolescents, which in turn informs programs aimed at improving adolescent health [13]. The purpose of this study was to determine the prevalence of suicidal attempts and examine associated factors among secondary school-going adolescents in Kilimanjaro region, northern Tanzania.

## 2. Methodology

### 2.1. Study Design and Settings

This was a cross-sectional study using secondary data from two repeated cross-sectional surveys in 2019 and 2022 years from the regional school health survey (RSHS) in the Kilimanjaro region, northern Tanzania. The surveys were conducted by the Institute of Public Health (IPH) of Kilimanjaro Christian Medical University College (KCMUCo) as part of a health promotion training activity for the Doctor of Medicine students at the college. The study was conducted in four districts of Kilimanjaro region, northern Tanzania, that is the Moshi municipality, Moshi rural, Hai, and Siha districts. Kilimanjaro is one of Tanzania’s 30 administrative regions, with an area of 1831.32 km^2^. Kilimanjaro has a population of 1.8 million people with 13.5% and 11.1% of them being adolescents aged 10–14 and 15–19 years, respectively [18]. Kilimanjaro has a relatively large number of secondary schools compared to other regions in Tanzania, with a total of 349 mixed and single-sex, day, and boarding schools. However, the population density of students per school is low, with an average of 413.2 students [19]. The methodology of the 2019 survey has also been described elsewhere [20].

### 2.2. Study Population and Sampling

The study population consisted of adolescents attending public secondary schools in the Kilimanjaro region who were present at school on the day of data collection. Adolescents with missing age or sex information and those with incomplete responses to the suicide questions were excluded from the analysis. Of the 4969 secondary school-going adolescents who participated in the survey, 4188 (84%) (3182 from Survey 1 and 1006 from Survey 2), aged between 13 and 17 years, were eligible and included in this analysis (Figure 1). A multistage sampling technique was used to select schools and students. In the first stage, four districts were purposely selected to ensure the representativeness of both rural and urban areas. In the second stage, a random selection of public secondary schools was made from all available schools in each district. In Survey 1, only form one students were selected, while in Survey 2, only form four students were selected to assess changes in adolescent risk behaviors three years after the initial survey in 2019 (The education system in Tanzania is based on a 7-4-2-3 model, which consists of 7 years of primary education, 4 years of secondary education (ordinary level), 2 years of advanced secondary education (advanced level), and 3 or more years of tertiary education). All available students were included in the interviews, which were conducted at the class level.

### 2.3. Data Collection

Data for Surveys 1 and 2 were collected using a self-administered questionnaire from the RSHS. The survey has been standardized to assess risk behaviors among school-going students in Tanzania and was administered in the Kiswahili language. The RSHS was adopted from the GSHS developed by WHO/CDC to help countries measure and assess behavioral risk and protective factors among students aged 13–17 years [21]. The standard English questionnaire from WHO was produced, translated into Kiswahili, pre-tested, and used in the Tanzania GSHS [22]. The collected data included basic demographic characteristics such as age, sex, and district, as well as information on violence, unintentional injury, dietary behaviors, hygiene, mental health, physical activity, substance use, and sexual behaviors. Before administering the questionnaires, the data collectors, who were trained medical students from KCMUCo, explained the study purpose to the students and answered any questions they had. The participants then filled out the questionnaires with guidance from the medical students. During data collection, the data collectors made the necessary efforts to preserve privacy and confidentiality. This was accomplished by clearly explaining its importance to participants and making sure there was some distance between them when filling out the questionnaires.

### 2.4. Instrument Measures

The outcome variable in this study was suicide attempt which was measured using a self-administered questionnaire from the RSHS [23]. Students in this study were asked, “During the past 12 months, how many times did you actually attempt suicide?”. Students were considered to attempt suicide if they attempted suicide at least once during the preceding 12 months. Suicidal plans and ideation were obtained from two questions “During the past 12 months, did you ever consider attempting suicide?” and “During the past 12 months, did you make a plan about how you would attempt suicide?” The response options were “yes or no”.

The explanatory variables included sociodemographic and behavioral characteristics, and social and psychological factors. Sociodemographic characteristics included: age in years, sex (male, female), and schooling district (Moshi municipality, Moshi rural, Siha, and Hai districts). Behavioral variables included ever being physically attacked (none, 1+ times); ever engaged in a physical fight (no, yes); ever being bullied (no, yes); currently using any substance ‘No’ if not used any of the substances (alcohol, cigarette, tobacco, and recreational drugs such as cocaine, heroin, marijuana, khat, and amphetamines) in the past 30 days and ‘Yes’ if otherwise; ever had sex (no, yes); and number of close friends (none, 1+ friend). Social and psychological variables were loneliness, food insecurity, lack of sleep, parental engagement, and support, (never, rarely, sometimes, most of the time; always), suicidal ideation (no, yes), and suicidal plan (no, yes).

### 2.5. Statistical Analysis

Data were cleaned and analyzed using SPSS software version 20. Categorical variables were summarized using frequencies and percentages whereas continuous variables were summarized using median and inter-quartile range. The chi-square test was used to compare suicidal attempt proportions by survey year and other participant characteristics. Multivariable logistic regression analysis was used to estimate odds ratios (OR) and 95% confidence intervals (CI) to determine factors associated with suicidal attempts at a 5% threshold level. The unadjusted logistic regression analysis determined the association between the independent variables and suicidal attempts (binary variable). For the adjusted logistic regression analysis, variables that had the likelihood ratio *p* values of <0.1 in the unadjusted analysis were included in the adjusted analysis. Age and sex were also included because they are known confounders of suicide attempts for adolescents.

## 3. Results

Among 4969 secondary school-going adolescents, 4188 (84%) (Survey 1, 3182; Survey 2, 1006) secondary school-going students aged between 13 and 17 years were eligible and included in this analysis (Figure 1). Overall, the median age was 15 years (IQR 14–16), while the mean age for Survey 1 was 14 years (SD 1.0), and the median age for Survey 2 was 17 years (IQR 16–17). More than half (53.0% and 64.8%) were females in surveys 1 and 2, respectively, and (41.6%; Survey 1 and 34.5%; Survey 2) were from Moshi district council (Table 1).

Participants in Survey 1 were less worried (*n* = 291, 9.4%) and lonely (*n* = 216, 6.8%) compared to those in Survey 2 (*n* = 161, 16% and *n* = 112, 11.1%, respectively). Over half (*n* = 409, 62.1%) of participants in Survey 1 were current substance users, while less than half in Survey 2 (*n* = 152, 32.5%) reported being current substance users. The proportion of participants who ever missed class was lower in Survey 1 (*n* = 607, 19.1%) than in Survey 2 (*n* = 289, 29.1%), while the proportion who ever experienced physical attack was higher in Survey 1 (*n* = 773, 24.3%) than in Survey 2 (*n* = 195, 19.6%). Participants in Survey 1 also reported more instances of ever engaging in a physical fight (*n* = 733, 23%) and ever bullying others (*n* = 434, 13.6%) compared to Survey 2 (*n* = 91, 9.1% and *n* = 63, 6.3%, respectively). Table 1 provides more details on these findings.

Overall, 137 (3.3%) secondary school-going adolescents reported having ever attempted suicide, 271 (6.5%) reported ever having suicidal ideation, and 178 (4.3%) reported ever making a suicidal plan. The reported number of suicide attempts was 95 (3.0%) in Survey 1 and 42 (4.2%) in Survey 2. The number of reported instances of suicidal ideation was 193 (6.1%) in Survey 1 and 78 (7.8%) in Survey 2, while the number of reported instances of making a suicidal plan was 127 (4.0%) in Survey 1 and 51 (5.1%) in Survey 2. The differences between the surveys were not statistically significant (Table 2).

In the adjusted analysis, overall, females had approximately three times higher odds (OR = 2.8, 95% CI 1.5–5.1) of attempting suicide compared to males. Participants who reported ever feeling lonely had two times higher odds (OR = 2.2, 95% CI 1.1–4.3), those who reported ever feeling worried had 99% higher odds (OR = 1.9, 95% CI 1.1–3.5), and those who reported ever being bullied had over two times higher odds (OR = 2.2, 95% CI 1.2–4.1) of attempting suicide compared to their counterparts. Gender (higher odds in females) (OR = 3.1, 95% CI: 1.5–6.5) was a significant factor for suicide attempts in Survey 1 but not in Survey 2. Ever having sexual experience (OR = 3.4, 95% CI: 1.1–10.7) was a significant factor for suicide attempts in Survey 2 but not in Survey 1 (Table 3).

## 4. Discussion

The study aimed to determine the prevalence of suicidal attempts and associated factors among secondary school-going adolescents in the Kilimanjaro region, northern Tanzania. In this study, the overall prevalence of suicidal attempts was 3.3%, suicidal ideation 6.5%, and suicidal plan 4.3%. Overall, suicidal attempts were associated with sex (higher odds in females), loneliness, worry, and bullying. Being female increased the risk for suicide attempts in Survey 1 but not in Survey 2, and ever had sex in Survey 2 but not in Survey 1.

In this study, the prevalence of suicidal attempts among secondary school-going adolescents in Kilimanjaro region, northern Tanzania was alarming. It is concerning to see a relatively high proportion of secondary school-going adolescents reporting having ever attempted suicide, experienced suicidal ideation, or made a suicide plan. This highlights the importance of addressing mental health issues among young people, especially in the school setting. This is a result of poor mental health, decision-making style, coping mechanism, family, and peer relationships [24]. The burden was slightly higher in Survey 2, which included older students (form fours, i.e., fourth and final year of ordinary level secondary school in Tanzania), compared to Survey 1, which included younger form one students. Older students are in late adolescence when they experience biological and social changes which increase the risk of mental health problems [14]. The fact that the differences between the two surveys were not statistically significant suggests that the issue of suicidal thoughts and behaviors is consistent across the study population. If this issue is not addressed, there will be more suicides, as reported by Makoye on the increased rate of suicide in Tanzania, especially among young people [25], which will have a long-term impact on the economy and mental health of the families, communities, and countries left behind by those who die by suicide. This prevalence is lower than the 11.5% reported among secondary school students in Tanzania’s 2014 GSHS report [13]. Weak cultural practices and taboos in Kilimanjaro compared to Dar es Salaam against suicide could be reflective of reporting bias, explaining these differences [26].

Suicide attempts were found to be associated with older adolescents in other studies [27], but not in this study. In addition, this study showed that the prevalence of suicide attempts was higher in females than in males, which is consistent with findings from studies in Jamaica [15] and Togo [28]. This could be because females are more likely to have mental health problems, as well as internalize emotional and behavioral problems than males, making suicide attempts more likely [8]. Furthermore, girls reach puberty earlier than boys due to pubertal changes and estradiol levels in girls, exposing them to love affairs that, if not successful, may lead to a suicide attempt as a coping mechanism for emotional problems [29]. Gender-based interventions that mentor girls by teaching them life skills and how to deal with adversity should be implemented in all schools.

Adolescent students who had ever felt lonely had a higher risk of suicidal attempts than their peers. School loneliness may be caused by a lack of desired peer and social relationships, belongingness, and having no one to share problems faced resulting in suicidal behaviors [30]. Lack of counseling services in schools or neglect may be a possible reason for loneliness, increasing the risk of suicide attempts [31].

Substance use was not associated with suicide attempts in this study but, adolescents who use substances have poor social and peer relationships which act as a driver for mental health problems that lead to suicidal behaviors, and have an impact on academic performance [32].

Secondary school-going adolescents who were ever worried were more likely to attempt suicide than those who were never worried. Worrying leads to self-doubt, insecurity, and confusion, which may lead to suicide attempts as a solution to anxiety or distress [30]. This has an impact on student performance in school because it reduces their ability to think or concentrate in class, which leads to school dropout, failure, and poor academic achievement [17]. In-school counseling programs for depressed students and survivors of suicide attempts should be established.

Bullying was also linked to suicide attempts in this study, which is like other studies in Mongolia [33] and from 90 countries in a pooled analysis [15]. Students who are victims of bullying have low self-esteem and experience loneliness, distress, and mental health problems [34]. Anti-bullying policies and programs in schools should be strengthened to prevent suicide in this population.

These results highlight the importance of addressing factors such as loneliness, bullying, and sexual experiences in preventing suicide among secondary school-going adolescents. It may also be important to consider gender differences in developing suicide prevention programs, as being female was a significant factor for suicide attempts in one of the surveys. Limiting access to suicide methods such as poisons, firearms, and drugs and treating mental disorders, media reporting on suicide, socio-emotional life skills in adolescents, and managing and following up on everyone affected by suicidal behaviors are all evidence-based interventions that can help prevent suicide attempts [34]. Given the high prevalence of suicide ideation and attempts, particularly among female students, healthcare professionals and educators should incorporate screening for suicidal thoughts and risk factors into routine care. Addressing factors such as loneliness, worry, and bullying in schools may also be effective in reducing the risk factors for suicidal attempts among adolescents. The differences in risk factors for suicidal attempts between the two surveys highlight the need for ongoing monitoring and targeted interventions to address the changing risk factors for suicide among adolescents.

### Limitations of the Study

Some limitations to this study should be noted. Because the survey only included school-going adolescents, the findings cannot be generalized to all adolescents in Tanzania and may underestimate the prevalence. The study relies on self-reported data and may not accurately reflect the true prevalence of suicidal thoughts and behaviors. It is also possible that some students may have underreported their experiences due to stigma or fear of consequences. This study’s repeated cross-sectional design did not link participants in the two surveys, potentially leading to duplicated responses, bias, and unaccounted overlap. This may have impacted the statistical power, representativeness, and generalizability of the findings. As such, interpreting the study’s results requires caution and future research should address overlap in serial surveys.

## 5. Conclusions

The present study suggests an alarmingly high level of suicidal attempts among secondary school-going adolescents in Kilimanjaro region, northern Tanzania with being female, ever having been lonely, ever having been worried, and ever being bullied being significant associated factors. Health promotion programs in schools should target these factors to prevent suicide attempts among adolescents.

## Figures and Tables

**Figure 1 behavsci-13-00288-f001:**
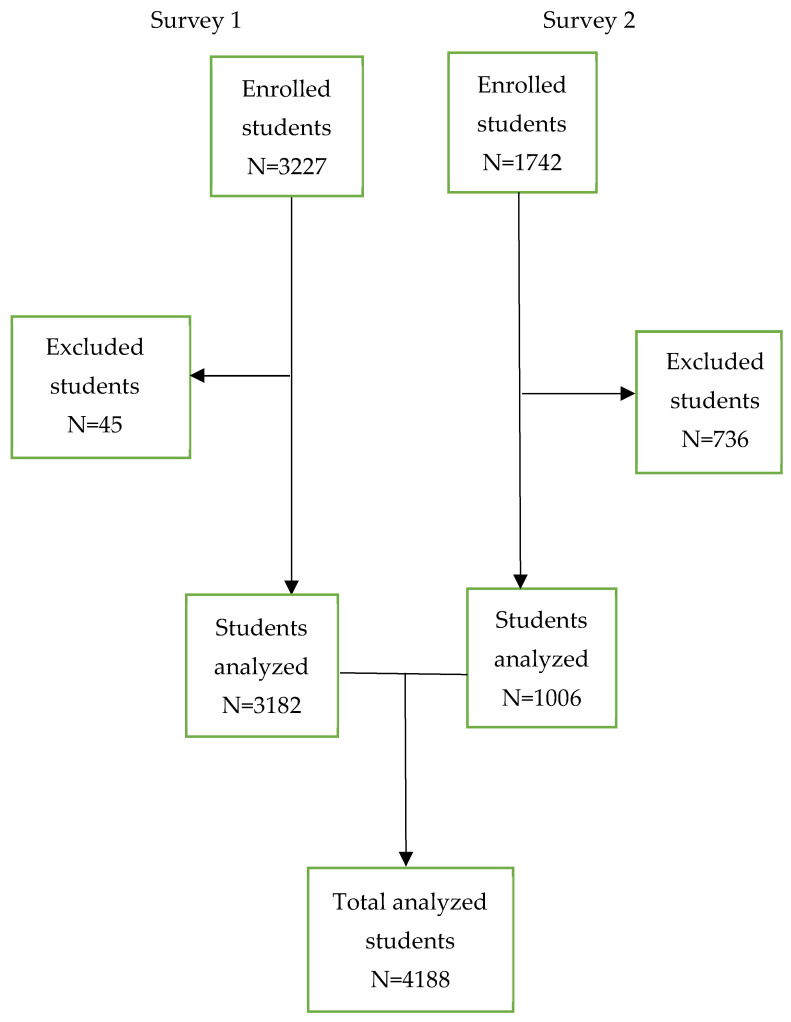
A flow diagram of enrolled and analyzed participants in the two surveys.

**Table 1 behavsci-13-00288-t001:** Sociodemographic and behavioral characteristics of secondary school-going adolescents from two repeated RSHS in 2019 and 2022 in Kilimanjaro region, northern Tanzania.

Variables	Overall *n* (%)	Survey 1 *n* (%)	Survey 2 *n* (%)	*p*-Value
**Age (years)**				<0.001
13–15	2733 (65.3)	2692 (84.6)	41 (4.1)	
16–17	1455 (34.7)	490 (15.4)	965 (95.9)	
Median (IQR), Mean (SD)	15 (14, 16)	14 (1.0)	17 (16, 17)	
**Sex**				<0.001
Male	1848 (44.1)	1494 (47.0)	354 (35.2)	
Female	2340 (55.9)	1688 (53.0)	652 (64.8)	
**Schooling district ***				<0.001
Moshi municipality	933 (22.5)	662 (20.8)	271 (28.0)	
Moshi district council	1658 (40.0)	1324 (41.6)	334 (34.5)	
Hai district council	646 (15.6)	523 (16.4)	123 (12.7)	
Siha district council	912 (22.0)	673 (21.2)	239 (24.7)	
**Ever been worried ***				<0.001
No	3638 (88.9)	2795 (90.6)	843 (84.0)	
Yes	452 (11.1)	291 (9.4)	161 (16.0)	
**Ever been lonely**				<0.001
No	3860 (92.2)	2966 (93.2)	894 (88.9)	
Yes	328 (7.8)	216 (6.8)	112 (11.1)	
**Current substances use * †**				<0.001
No	566 (50.2)	250 (37.9)	316 (67.5)	
Yes	561 (49.8)	409 (62.1)	152 (32.5)	
**Ever had sex ***				0.399
No	3784 (90.5)	2886 (90.7)	898 (89.8)	
Yes	398 (9.5)	296 (9.3)	102 (10.2)	
**Ever missed class ***				<0.001
No	3277 (78.5)	2571 (80.9)	703 (70.9)	
Yes	896 (21.5)	607 (19.1)	289 (29.1)	
**Number of close friends ***				0.290
No friends	363 (8.7)	285 (9.0)	78 (7.9)	
>1 friend	3798 (91.3)	2888 (91.0)	910 (92.1)	
**Ever been physically attacked ***				0.002
No	3208 (76.8)	2408 (75.7)	800 (80.4)	
Yes	968 (23.2)	773 (24.3)	195 (19.6)	
**Ever engaged in a physical fight ***				<0.001
No	3363 (80.3)	2449 (77.0)	914 (90.9)	
Yes	824 (19.7)	733 (23.0)	91 (9.1)	
**Ever been bullied ***				<0.001
No	3688 (88.1)	2748 (86.4)	940 (93.7)	
Yes	497 (11.9)	434 (13.6)	63 (6.3)	
**Food insecurity ***				<0.001
No	3885 (92.8)	2924 (91.9)	961 (95.7)	
Yes	301 (7.2)	258 (8.1)	43 (4.3)	
**Parental engagement ***				0.058
No	2156 (51.6)	1666 (52.4)	490 (49.0)	
Yes	2025 (48.4)	1514 (47.6)	511 (51.0)	
**Survey year (Overall)**	4188	3182 (76.0)	1006 (24.0)	

IQR = Interquartile range. * Frequency does not tally due to missing values. † Using at least one substance (alcohol, cigarette, tobacco, khat, marijuana, and amphetamines) in the past 30 days.

**Table 2 behavsci-13-00288-t002:** Prevalence of suicidal attempts by survey year among secondary school-going adolescents in Kilimanjaro region, northern Tanzania.

Variable	Overall *n* (%)	Survey 1 *n* (%)	Survey 2 *n* (%)	*p*-Value
**Ever attempted suicide**				
No	4051 (96.7)	3087 (97.0)	964 (95.8)	0.065
Yes	137 (3.3)	95 (3.0)	42 (4.2)	
**Ever had suicidal Ideation ***				
No	3912 (93.5)	2989 (93.9)	923 (92.2)	0.053
Yes	271 (6.5)	193 (6.1)	78 (7.8)	
**Ever made suicidal Plan ***				
**No**	4007 (95.7)	3055 (96.0)	952 (94.9)	0.135
**Yes**	178 (4.3)	127 (4.0)	51 (5.1)	

Overall N = 4188, Survey 1, N = 3182, Survey 2, N = 1006, * Frequency does not tally due to missing values.

**Table 3 behavsci-13-00288-t003:** Analysis of factors associated with suicidal attempts among secondary school-going adolescents from two repeated RSHS in 2019 and 2022 in Kilimanjaro region, northern Tanzania.

Variables	Overall		Survey 1		Survey 2	
	AOR (95% CI)	*p*-Value	AOR (95% CI)	*p*-Value	AOR (95% CI)	*p*-Value
**Age (years)**						
13–15	1		1		1	
16–17	1.0 (0.6, 1.7)	0.924	1.6 (0.7, 3.9)	0.277	0.7 (0.1, 8.0)	0.775
**Sex**						
Male	1		1		1	
Female	2.8 (1.5, 5.1)	0.001	3.1 (1.5, 6.5)	0.002	2.9 (0.9, 9.3)	0.078
**Ever felt lonely (Yes)**	2.2 (1.1, 4.3)	0.020	2.4 (1.0, 5.7)	0.055	1.9 (0.7, 5.5)	0.215
**Ever been worried (Yes)**	1.9 (1.1, 3.5)	0.033	2.0 (0.9, 4.3)	0.087	2.2 (0.8, 6.1)	0.112
**Current substance use (Yes)**	1.8 (1.0, 3.3)	0.058	1.7 (0.7, 3.7)	0.230	1.8 (0.7, 4.5)	0.218
**Ever had sex (Yes)**	1.5 (0.8, 2.9)	0.235	1.0 (0.4, 2.3)	0.901	3.4 (1.1, 10.7)	0.037
**Ever physically attacked (Yes)**	1.1 (0.6, 1.9)	0.789	0.9 (0.4, 1.8)	0.708	1.6 (0.6, 4.2)	0.360
**Ever engaged in a physical fight (Yes)**	1.2 (0.6, 2.2)	0.590	1.2 (0.6, 2.5)	0.579	1.0 (0.3, 3.4)	0.955
**Ever bullied (Yes)**	2.2 (1.2, 4.1)	0.016	2.1 (1.0, 4.5)	0.056	2.4 (0.8, 7.7)	0.130
**Parental engagement (Yes)**	1.2 (0.7, 2.1)	0.454	1.3 (0.7, 2.6)	0.451	1.2 (0.5, 2.9)	0.709

AOR—Adjusted Odds Ratio. Overall N = 4188, Survey 1, N = 3182, Survey 2, N = 1006.

## Data Availability

The data presented in this study are available on request to the Director of the Institute of Public Health at KCMUCo at iph@kcmuco.ac.tz or through the corresponding author.

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
