# Peer review of "Suicidal Attempts among Secondary School-Going Adolescents in Kilimanjaro Region, Northern Tanzania"

_behavsci, 2023, doi:10.3390/bs13040288_

Round 1
Reviewer 1 Report
I found this a very interesting study as not many studies regarding attempted suicide are available from African countries, and Tanzania specifically. Despite the merits of the study, there are also a few comments, listed below.
Introduction
The WHO has estimated that about 700,000 people die by suicide each year, globally. Then how can 600,000 children and adolescents die by suicide annually?
Avoid the expression “commit” suicide, as this may refer to a criminal act, and increases stigmatization of suicide. Maybe use “die by suicide” or similar.
Please check throughout.
Incorrect sentence: “Increased suicide rates among adolescents 15-19 years have been seen to increase with rate of 0.64 per 100000”
Incorrect sentence: “If suicidal attempts are not addressed promptly, they lead to suicide”.
You could say that attempted suicide is a major risk factor for dying by suicide, but you cannot say that it ‘leads’ to suicide.
Methods
What are form one and form four students?
How many students of survey 1 also participated in survey 2?
How was the overlap between the samples accounted for in the analysis?
In the analysis, they are treated as two separate groups. Are you sure that there is no overlap in participants?
Did participants receive a definition of attempted suicide?
Discussion
Could you add a paragraph with clinical implications (e.g., for support and service delivery), and further research.
Good luck with revising the manuscript.
Reviewer 2 Report
Suicidal attempts among secondary school-going adolescents Kilimanjaro region, northern Tanzania
This is an important contribution to the field of suicidal behavioral research. The article is well written but methodologically fragile, since it relies its measures on one single question asked to school-aged participants. Despite this, if the editor considers the submission, I believe a several changes would improve its overall chances of being published:
1. The introduction section is very brief and hardly describes the state of the art on this topic, especially on psychosocial determinants of this behavior. Also, focus on aspects of suicidal behavior (behavior, ideation, intention) should be defined.
2. Objectives must be clearly and thoroughly indicated.
3. Line 57-60 should be in the discussion section.
4. Participants in this study were underaged. How was the informed consent obtained?
5. An ethical considerations section should be included, stating what was done to prevent adolescent to feel vulnerable after participating in a study about suicide attempts.
6. The SPSS version authors used is far outdated (v.20), the current version is 29.
7. The study variables section should be replaced with instrument measures.
8. Also, the lack of psychometrically validated measures poses important vulnerabilities to the study. Please discuss this.
9. Authors mention, in the study variables section, that only one question was asked (During the past 12 months, how many times did you actually attempt suicide?), but in the results section, they present results on attempt, ideation, and planning. Please clarify.
10. An implications section should be included in the discussion section, focusing on mental health and educational policies to prevent suicide in school-aged individuals in northern Tanzania.
Best wishes.
Reviewer 3 Report
This manuscript describes a cross-sectional study involving suicide attempts among adolescents in Tanzania. The literature review is brief, yet appropriate since the discussion includes quite a few connections to other publications. The methodology is appropriate to the research question. The results are presented clearly. As already mentioned, the discussion is thorough, offering connections to previous publications.
The value of the manuscript is high, as adolescent lives are involved. Bringing more attention and awareness to this issue could result in fewer adolescent suicides. Overall, it is well written. Recommendation is to accept for publication in present form.
Round 2
Reviewer 2 Report
Thank you for implementing all the requested changes. I believe the article is now fit for publication.
Best wishes.